# Alzheimer's scRNA-seq Data Analysis Using Multi-type Deep Autoencoders

## Abstract

Single-cell RNA sequencing (scRNA-seq) technology has been applied in Alzheimer's disease (AD) research to explore its pathogenic mechanism. The complexity of analyzing and utilizing sequencing data is significantly amplified by the high dimensionality and high noise levels of the data, as well as the presence of missing data. In order to tackle these problems, we suggest implementing a novel data processing framework that consists of two primary algorithms: the imputation algorithm scICLGAE and the clustering algorithm scCapsZB. scICL-GAE employs two graph autoencoders (GAE) to fill in missing values by utilizing comparison learning to filter similar nodes from both global information and local structure. In order to verify the imputation impact and enhance the precision of the clustering outcomes, we have devised the scCapsZB algorithm. scCapsZB is a method that integrates a capsule network and a zero-inflated negative binomial distribution (ZINB) autoencoder. It incorporates prior knowledge through the capsule network and employs a self-attention routing mechanism to reduce the number of training parameters. Additionally, it uses the ZINB model to capture the feature representation of the data. The testing of our new framework on both generic and Alzheimer's datasets demonstrates substantial enhancements.

## 1 Introduction

Single-cell RNA sequencing (scRNA-seq) technology enables the understanding of the potential mechanisms of diseases through dynamic gene expression analysis Zhang et al. (2023) and has emerged as a transformative instrument for research in the fields of biology and medicine Luo et al. (2025). It significantly enhances the identification of a range of diseases, including cancer Huang et al. (2023), immune system abnormalities Yin et al. (2024), mental illnesses Zhou et al. (2023), and chronic diseases Nakayama et al. (2024). Recently, researchers have applied scRNA-seq to Alzheimer's disease (AD) prevention and treatment Cisterna-García et al. (2023). By the year 2050, it is projected that the number of individuals diagnosed with Alzheimer's disease will surpass 150 million, thereby exerting a considerable impact on both familial structures and public health systems Huang et al. (2024). The scRNA-seq technology facilitates the investigation of neural development associated with Alzheimer's disease and enables the examination of cellular alterations occurring prior to and following the onset of the disease Feng et al. (2024). Nonetheless, the high dimensionality, sparsity, and inherent noise associated with sequencing data pose significant challenges to data analysis within the context of Alzheimer's disease research Kharchenko (2021).

Recent advancements in imputation algorithms have been made to enhance the quality of scRNA-seq data through the integration of autoencoders (AE) with deep neural networks (DNN), such as DCA Eraslan et al. (2019), SAVER-X Wang et al. (2019), GraceImpute Wang et al. (2025), and scGMAI Yu et al. (2021). For the limitations of AE, these improved algorithms have proven effective. DeepImpute Arisdakessian et al. (2019) employs a divide-and-conquer strategy within a DNN model framework to forecast gene expression levels. scIGANs Xu et al. (2020), which are founded on the principles of generative adversarial network (GAN), address the issue of over-smoothing by producing synthetic cellular data. This approach effectively enhances performance consistency across various cell populations. Clustering algorithms such as Dhaka Rashid et al. (2021), scvis Ding et al. (2018), and scVAE Grønbech et al. (2020) primarily combine DNN with variational autoencoder (VAE). Another effective method, scDeepCluster Tian et al. (2019), simultaneously learns feature representations while explicitly modeling cell clusters. Additionally, models such as scVI,

LDVAE Svensson et al. (2020), SAUCIE Amodio et al. (2019), and scScope Deng et al. (2019) integrate AE and VAE with multiple analytical functions. However, these unsupervised methodologies frequently yield outcomes that are devoid of biological relevance and tend to place disproportionate emphasis on gene attributes, thereby overlooking the interrelationships among cellular genes.

In this paper, we introduce a new data processing framework that includes imputation algorithm scICLGAE and clustering algorithm scCapsZB to tackle these issues. The imputation uses two graph autoencoders (GAEs) for contrastive learning, which combines local and global information to find similar nodes.For clustering, we use zero-inflated negative binomial distribution (ZINB) autoencoder and capsule network for feature extraction and denoising to capture gene relationships, coordinated by supervisory module. Our model outperformed others across 12 scRNA-seq datasets. When applied to AD data, improved clustering revealed changes in disease-related cell proportions, providing valuable insights for prevention and treatment.

## 2 METHODS

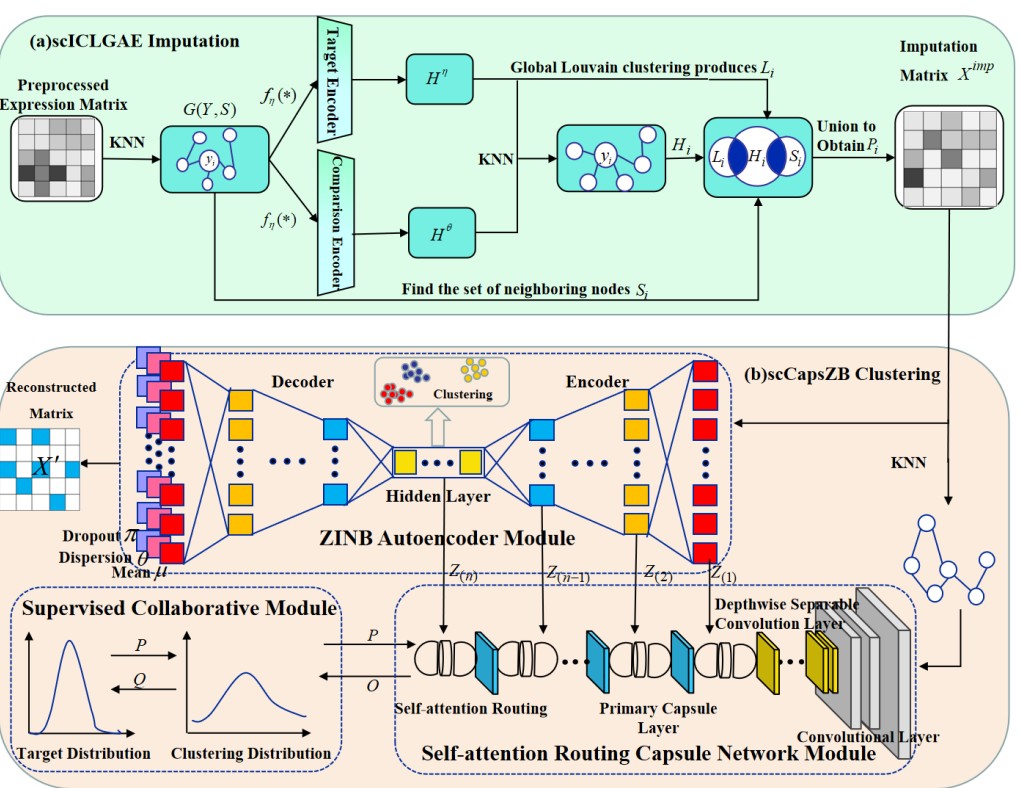

Figure 1: Model framework: The preprocessed expression matrix is transformed into a KNN graph via cosine similarity and fed into a contrastive encoder for new node representations. A recalculated KNN graph helps identify nodes similar to the target for data imputation. The imputed matrix is re-analyzed to obtain a new KNN graph, which serves as input for ZINB autoencoder and self-attention routing capsule network in cell clustering, with the overall process optimized by a supervised collaborative module.

### 2.0.1 CONSTRUCTING CELL MAPS.

We use the KNN algorithm to construct a cell graph, where nodes represent cells and edges represent connections between them. In this graph construction, the $k$ value of the KNN algorithm is utilized to control the scale of adjacent nodes and measure the proximity of distances. Specifically, nodes within the $k$ shortest distances from a given node are regarded as its neighbors and connected. To incorporate more potentially related cells in the analysis, a relatively larger $k$ value is set. The

weights of these edges are calculated using cosine similarity, as shown in Equation (2):

$$S_{a,b} = \frac{\sum_{a=1,b=1}^{n} Z_a \cdot Z_b}{\sqrt{\sum_{a=1}^{n} Z_a^2}\sqrt{\sum_{b=1}^{n} Z_b^2}} \tag{1}$$

where $Z_a$ and $Z_b$ represent cells $a$ and $b$, respectively. $S_{a,b}$ is the cosine similarity between these two nodes and its constituent matrix $S$, which is also the adjacency matrix that records the association information, while considering $Y$ as its cell node matrix, the constructed KNN graph can be denoted as $G(Y, S)$.

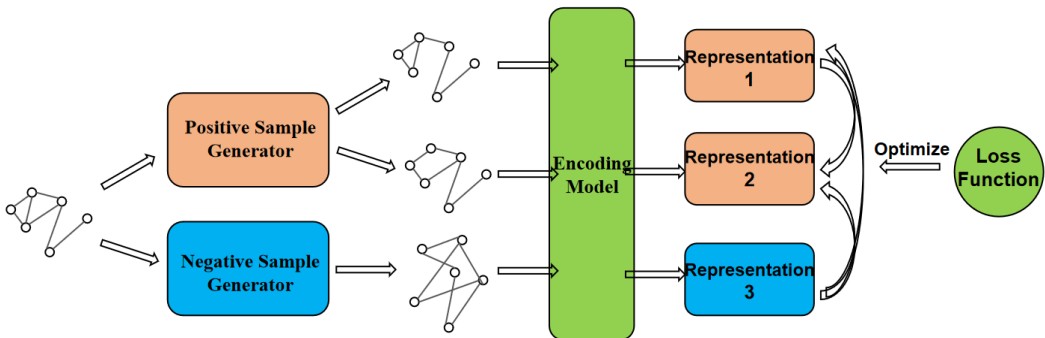

Figure 2: Graph contrastive learning framework: It consists of a positive sample generator, a negative sample generator, a encoding model, and a loss function. The positive and negative sample generators input the same graph. The encoding model learns the samples output by the generators and gets the corresponding feature representations. Finally, the parameters are trained by maximizing the difference of the positive and negative feature representations.

### 2.0.2 CONTRAST LEARNING NODAL IMPUTATION.

The graph $G$ is provided as input to the target encoder $f_\eta(*)$ and control encoder $f_\theta(*)$. Fig. 2 shows the graph contrastive learning framework Cen et al. (2023). These two autoencoders employ graph convolutional network (GCN) internally and generate corresponding node representations for each cell:

$$H^\eta = f_\eta(Y, S) \tag{2}$$

$$H^\theta = f_\theta(Y, S) \tag{3}$$

where $H^\eta$ and $H^\theta$ is a summary of the individual cell representation. In order to get the whole set of sample nodes $H_i$ that are similar to the target cell node $y_i$, it is necessary to recalculate the cosine similarity distance for both positive and negative samples:

$$sim\,(y_i, y_j) = \frac{h_i^\theta \cdot h_j^\eta}{\parallel h_i^\theta \parallel \parallel h_j^\eta \parallel}, \forall y_i \in Y \tag{4}$$

where $h_i^\theta$ represents the embedded output of the control encoder for cell $y_i$, and $h_j^\eta$ represents the embedded output of the target encoder for the other cell $y_j$. By applying KNN with a smaller $k$ value than that in cell graph construction (using a smaller value is to screen similar cells), we obtain $H_i$.

To get the set of locally most similar samples, we first define $S_i$ as the collection of nodes that have a direct connection with the target cell $y_i$ in the edge weight matrix $S$. Then, the set of locally most similar samples is achieved by computing the intersection of $H_i \cap S_i$.

To globally find other nodes similar to the target encoder embedding $h_i^\eta$, we apply the Louvain De Meo et al. (2011) algorithm to get the set $L_i$. The globally most similar set is then obtained by intersecting $H_i$ with this globally-derived $L_i$. Given that $L_i$ is based on positive samples, this intersection serves to enhance the positive sample representation. Finally take the union set:

$$P_i = (H_i \cap S_i) \cup (H_i \cap L_i) \tag{5}$$

For each cell $i$, if the expression of gene is not equal to zero, retain its value. If not, the imputed value should be the average expression of gene $j$ from all cells $p_k$ in the same node set $P_i$. The calculating process for the gene deletion value in cell $i$ is illustrated by Equation (7), resulting in the estimated gene expression matrix $X^{imp}$.

$$
\begin{cases}
X_{i,j} & if\, X_{i,j} > 0 \\
\frac{\sum_{p_k \in P_i} p_{i,j}}{Count(P_i)} & if\, else
\end{cases}
\tag{6}
$$

The contrastive loss function minimizes the cosine similarity distance of node representations from the target and contrastive autoencoders, enhancing their similarity during training and stabilizing the resultant similar node set. The loss function is shown in Equation (8).

$$
L_{con} = -\frac{1}{N} \sum_{i=1}^{N} \sum_{p_k \in P_i} \frac{f_\theta(y_i, S) f_\eta(p_k, S)^T}{\|f_\theta(y_i, S)\| \|f_\eta(p_k, S)\|}
\tag{7}
$$

## 2.1 CLUSTER ANALYSIS

We enhance scCapsZB by using sub-labeled data for semi-supervised learning. The GNN is replaced with a capsule network connected to the ZINB autoencoder for cooptimization. Supervised module trains the entire network uniformly for clustering. See Fig.1(b) for the structure.

### 2.1.1 ZINB AUTOENCODER MODULES.

The processed gene expression matrix is fed into the ZINB autoencoder, which consists of three parts: the encoder, the hidden layer, and the decoder. Gene features are embedded into a low-dimensional space using the similarity between the ZINB distribution and scRNA-seq data. The distribution functions of Negative Binomial (NB) and ZINB are shown in the followings.

$$
NB(X; \mu, \theta) = \frac{\Gamma(X + \theta)}{\Gamma(\theta)} \left(\frac{\theta}{\theta + \mu}\right)^\theta \left(\frac{\mu}{\theta + \mu}\right)^X
\tag{8}
$$

$$
ZINB(X; \pi, \mu, \theta) = \pi \delta_0(X) + (1 - \pi) NB(X; \mu, \theta)
\tag{9}
$$

where $X$ is preprocessed matrix. $\mu$, $\theta$ and $\pi$ represent mean, variance, and dropout rate respectively, which are estimated by connecting three independent fully connected layers to the last layer of the decoder.

The encoder converts the preprocessing matrix $X$ into the feature representation $Z$ of the intermediate hidden layer using the following equation:

$$
Z = f_{enc}(WX + b)
\tag{10}
$$

where $W$ is the encoder weight vector; $b$ is the encoder offset; $f_{enc}$ represents the encoder function of the abstract representation. If the encoder has $k$ layers, the learning process is as follows:

$$
Z_{(k)} = \phi\left(w_{(k)} Z_{(k-1)} + b_{(k)}\right)
\tag{11}
$$

where $Z_{(k)}$ is the representation of the features learned in the $k$-th layer, $w_{(k)}$ is the weight of the layer, and $b_{(k)}$ is the offset term. The decoder transforms the feature representation $Z$ from the intermediate hidden layer into the output matrix $X'$ using the following process:

$$
X' = f_{dec}(W'Z + b')
\tag{12}
$$

where $W'$ is the decoder weight vector; $b'$ is the decoder offset; $f_{dec}$ represents the decoder function of the abstract representation.

The loss function is defined as the sum of the negative logarithms of the ZINB distribution:

$$
L_{zinb} = \sum -\log(ZINB(X|\pi, \mu, \theta))
\tag{13}
$$

The decoder's final layer output is given by the following equation:

$$
D = f_{dec}(f_{enc}(X))
\tag{14}
$$

### 2.1.2 Self-attention Routing Capsule Network Module.

The module has four components: convolutional, depthwise separable convolutional, primary capsule and self-attention routing. For visualizing cell topology, a KNN graph is built on the processed gene expression matrix using Pearson correlation. The first two parts, like a regular convolutional network, use ordinary convolution and batch normalization for feature extraction and high-dim projection. Unlike traditional capsule nets Duarte et al. (2021); Chen et al. (2025); **?**, a depthwise separable convolution Balmez et al. (2025) is added in the second part to reduce capsule creation params. The last two parts are interleaved.

At the primary capsule layer, to make vector length denote entity probability and enable high-level capsule to predict its parameters from low-level output, a new activation function is used:

$$O_n^i = \text{squash}\left(g_n^i\right) = \left(1 - \frac{1}{e^{\|g_n^i\|}}\right)\frac{g_n^i}{\|g_n^i\|} \tag{15}$$

where $g_n^i$ represent the input of the $n$-th capsule in the $i$-th layer, and $O_n^i$ is the output vector with the same dimension and characteristics as $g_n^i$.

In this module, self-attention routing Mazzia et al. (2021) replaces traditional dynamic routing. It enables the output vectors of active capsules to reach corresponding high-level capsules. The input $g_n^{i+1}$ of a high-level capsule is the weighted sum of the prediction vectors from the lower-level capsule $O_n^i$. When obtaining $g_n^{i+1}$, we incorporate the prior probability matrix $L^i$ into the coupling coefficients $C^i$ derived from the self-attention tensors $E^i$ to get the self-attention routing weights. We integrate the knowledge from the ZINB autoencoder with the capsule network's outputs to optimize both modules:

$$O_n^i = \sigma O_n^i + (1 - \sigma) Z_{(i)} \tag{16}$$

Capsule network loss has two parts: sorting loss and KNN graph reconstruction loss, the former being the sum of losses by all digital capsule layers:

$$L_{clu} = T_{n^L} max(0, m^+ - \|O_n^L\|)^2 + \lambda (1 - T_{n^L}) max(0, \|O_n^L\| - m^-)^2 \tag{17}$$

where $O_n^L$ is the final layer's capsules. $m^+$, $m^-$ and $\lambda$ are hyperparameters. If the entity corresponding to the capsule exists, $T_{n^L}$ is 1; otherwise, it's 0.

The reconstruction loss is defined by the Euclidean distance between the reconstructed and input graphs:

$$L_{recon} = \text{dist}\left(G, \hat{G}\right) = \sqrt{\sum_{i=1}^{n}\left(G_i - \hat{G}_i\right)^2} \tag{18}$$

where $G$ is the input graph matrix, and $\hat{G}$ is the reconstructed graph matrix.

For semi-supervised network training and prior knowledge integration in capsule network clustering, this module uses a few real cell type labels instead of golden ones. Divide the dataset into sub-data and sub-label data with a certain ratio, ensuring much more sub-data. Mark real cell type numbers on sub-label data cells for accurate clustering while leaving sub-data cells unmarked. In each training round, input the marked sub-label data to initialize with its real cell types and calculate its loss.

$$L_{\text{sub-label}} = L_{\text{clu}}^{\text{sub-label}} + L_{recon}^{\text{sub-label}} \tag{19}$$

Then input the unlabeled sub-data. As it lacks labels, only the reconstruction graph loss can be computed. But with the prior sub-label data and its labels, the sub-data can be classified and its classification loss adopts the sub-label data loss. The total loss of the capsule network is shown as follows.

$$L_{\text{cap}} = L_{\text{clu}}^{\text{sub-label}} + L_{\text{sub-label}} \tag{20}$$

### 2.1.3 Supervised Collaborative Module.

This module trains the entire network through the target distribution $P$, the clustering distribution $Q$, and the probability distribution $O$. $Q$ use the Student's distribution to model the probability of

all cells being assigned to a K-means clustering center. The $q_{it}$ element quantifies the similarity between the data representation $z_i$ of the ZINB encoder layer $i$ and the vector representation $\mu_t$ of the clustering center $t$:

$$q_{it} = \frac{\left(1 + \|\mu_t - z_i\|^2 / f\right)^{-\frac{f+1}{2}}}{\sum_i (1 + \|\mu_t - z_i\|^2 / f)^{-\frac{f+1}{2}}} \tag{21}$$

where $f$ is the degree of freedom. The component $p_{it}$ of $P$ is a more reliable representation of the data generated using $q_{it}$, which is the actual clustering center of the data. This computation is performed according to the Equation (23).

$$p_{it} = \frac{q_{it}^2 / \sum_t q_{it}}{\sum_{t'} (q_{it'}^2 / \sum_{t'} q_{it'})} \tag{22}$$

Clustering minimizes the log cross-entropy of target distribution $P$ and clustering distribution $Q$ to match $Q$'s centers with the data's true centers, maximizing cluster compactness and separation. The loss is shown in Equation (24):

$$L_{cluster} = -p_{\text{it}} \log(q_{\text{it}}) - (1 - p_{\text{it}}) \log(1 - q_{\text{it}}) \tag{23}$$

$P$ is computed from $Q$, meaning $Q$ guides $P$'s learning. From the capsule network, we get $o_{it}$ which holds prior knowledge and cell details, representing the probability of cell $i$ in cluster $t$. Using KL divergence Cui et al. (2023), we calculate the loss for $P$ to supervise $O$, with the loss function in Equation (25).

$$L_k = \text{KL}(\text{P}\|\text{O}) = \sum_i \sum_t p_{it} \log \frac{p_{it}}{o_{it}} \tag{24}$$

| Dataset | Sequencing Platform | Cell Number | Gene Number | cell categories |
|---|---|---|---|---|
| 10X_PBMCZheng et al. (2017) | 10X | 4271 | 16499 | 8 |
| RomanovRomanov et al. (2017) | Smart-seq2 | 2881 | 24341 | 7 |
| Human1Baron et al. (2016) | inDrop | 1937 | 20125 | 14 |
| Human2Baron et al. (2016) | inDrop | 1724 | 20125 | 14 |
| Human3Baron et al. (2016) | inDrop | 3605 | 20125 | 14 |
| Human4Baron et al. (2016) | inDrop | 1303 | 20125 | 14 |
| Mouse1Baron et al. (2016) | inDrop | 822 | 14878 | 13 |
| Mouse2Baron et al. (2016) | inDrop | 1064 | 14878 | 13 |
| CITE_CMBCMimitou et al. (2019) | 10X | 8617 | 2000 | 15 |
| ZeiselZeisel et al. (2015) | Drop-seq | 3005 | 19972 | 9 |
| KleinKlein et al. (2015) | inDrop | 2717 | 24175 | 4 |
| Human_kidneyYoung et al. (2018) | 10X | 5685 | 33658 | 11 |

Table 1: Statistics of processed datasets.

## 3 EXPERIMENTS

### 3.1 EXPERIMENTAL SETTINGS

#### 3.1.1 DATASETS.

We evaluated our model using 12 scRNA-seq datasets from sequencing platforms, as shown in Table 1. Each dataset contains between 1000 and 9000 cells, all genetically annotated with known cell types. In the imputation section, we added a supplementary dataset, GSE138852 Grubman et al. (2019) which includes 6 Alzheimer's patients and 6 healthy controls, totaling 13,214 cells (6,541 control, 6,673 AD) and 10,850 genes across 8 cell types or subtypes.

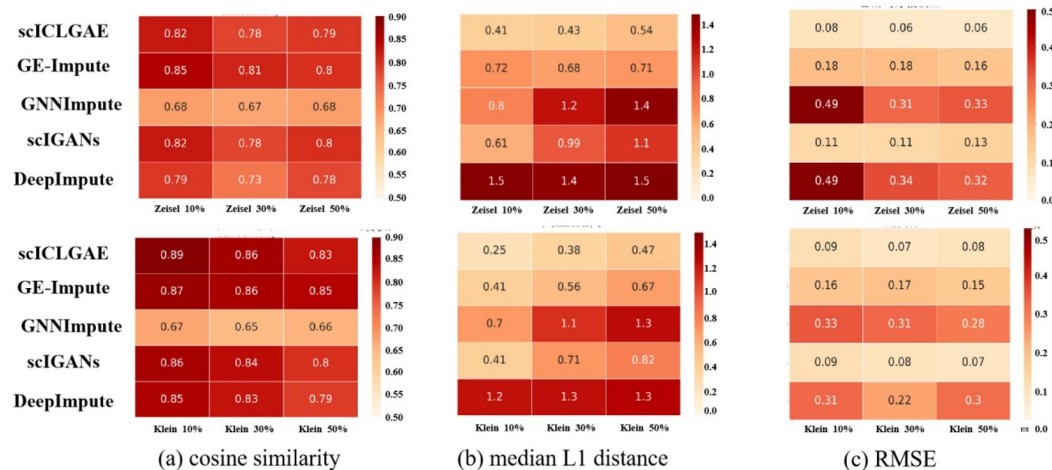

Figure 3: Comparison and evaluation of imputation metrics.

### 3.1.2 BASELINE.

The algorithm's performance is assessed against various advanced imputation and clustering methods for scRNA-seq data. DeepImpute Wang et al. (2021b) utilizes sub-neural networks with a dropout layer for gene imputation. scIGANs Zheng et al. (2017) and GE-Impute Wu & Zhou (2022) apply adversarial and neural network models to fill missing values. GNNImpute Xu et al. (2021) and scCDG Wang et al. (2021a) leverage graph attention convolution and GNNs to consolidate similar cell information, reduce noise, and transform high-dimensional data into low-dimensional representations. scClust Chorbadjiev et al. (2020) implements hierarchical clustering with predetermined limits, while PARC Stassen et al. (2020) and graph-sc Ciortan & Defrance (2022) enhance processing speed through graph pruning and feature extraction using graph autoencoders. DCAkmeans Eraslan et al. (2019) joins AE noise reduction with K-means clustering, and scDeepCluster Tian et al. (2019) minimizes loss using a denoising AE and ZINB model. Seurat Satija et al. (2015) adopts the Louvain method for clustering.

### 3.1.3 EVALUATION METRICS.

To evaluate the imputation effect of scICLGAE and the clustering performance of scCapsZB, we selected three common similarity metrics: cosine similarity, median L1 distance, root mean square error (RMSE) and two clustering evaluation indicators: normalized mutual information (NMI) and adjusted Rand index (ARI).

## 3.2 RESULTS

### 3.2.1 COMPARISON OF IMPUTATION METRICS:

To assess scICLGAE's imputation performance, we used the Zeisel and Klein datasets with gold-standard cell type labels. Following scVI's Leave-one-out strategy Chen et al. (2024), dropout events were simulated via Splatter Zappia et al. (2017) by randomly zeroing some non-zero gene expressions. Three missing rates (10%, 30%, 50%) were set. We evaluated scICLGAE against four methods (DeepImpute, scIGANs, GNNImpute, GE-Impute) using Cosine Similarity, Median L1 Distance, and RMSE between original and imputed gene expression matrices. Results are in Fig. 3.

scICLGAE shows comparable performance in cosine similarity to recent methods like those based on deep neural, generative adversarial, and graph neural networks. It attains optimal or near-optimal results on most datasets (excluding the Zeisel dataset with 50% missing rate). For L1 distance and RMSE, it excels except on the Klein dataset with 50% missing rate, with more significant improvement. This is because cosine similarity emphasizes vector direction, while L1 and RMSE focus on vector attributes, better reflecting the preservation of original cell expression. Overall, compared with recent deep learning imputation methods, scICLGAE has made notable progress and

can effectively impute missing values in the original gene expression matrix, facilitating subsequent analyses.

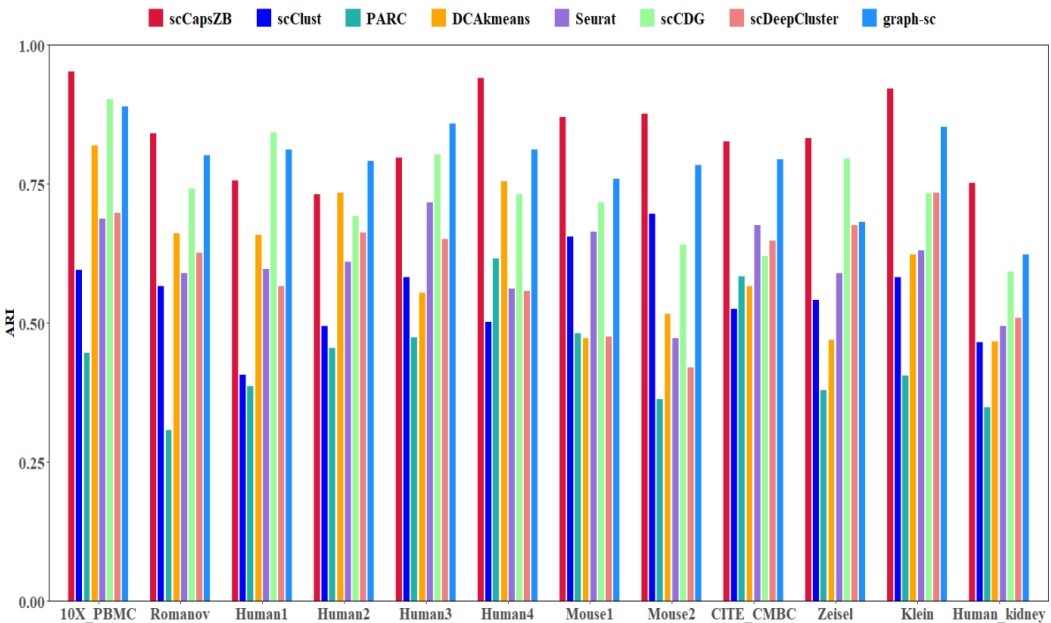

Figure 4: Comparison of ARI metrics.

### 3.2.2 COMPARISON OF CLUSTERING METRICS:

To evaluate scCapsZB's clustering, it was tested on 12 datasets with 7 other algorithms, using ARI and NMI. Among 9 datasets, scCapsZB led in ARI, surpassing neural network-based ones by 5.59% - 23.22% and statistical learning-based ones by up to 173.95%, as shown in the Fig. 5. Its ARI exceeded 0.9 on some datasets. Even ranking third on a few datasets, the gaps were minor. Performance on certain datasets may be due to training data or ZINB. The results shown in the NMI values were the same as those in the ARI.

Clustering metric comparison reveals scCapsZB's superiority over methods ignoring cell relations and topology, though its edge over advanced graph neural methods is subtle. The Romanov dataset (where scCapsZB's ARI is close to graph - sc and scCDG) was chosen. Their Embeddings were UMAP-projected for visualization, as shown in Fig. 6. scCapsZB shows good clustering with clear separation and compactness, highlighting its techniques' merits. graph - sc has one less cluster, leading to cell misclassification and biological misalignment despite decent overall metrics. sc-CDG's cluster number is correct, but some clusters crowd due to potential over-denoising, losing data heterogeneity and affecting dimensionality reduction.

### 3.2.3 IMPUTATION AND CLUSTERING ANALYSIS OF AD SEQUENCING DATA:

To investigate cellular changes before and after Alzheimer's disease (AD) onset and assess the performance of scICLGAE and scCapsZB on large-scale single-cell RNA-seq data, we first imputed the GSE138852 dataset using scICLGAE and then clustered the results with scCapsZB. As shown in Fig. 7(a), eight cell clusters were identified, with the right panel illustrating their distribution in control and AD samples. Unlike standard datasets, GSE138852 shows cluster adhesion and reduced separability, likely due to glial cell characteristics. However, the combination of imputation and capsule network prior enables biologically meaningful clustering; for instance, Oligo_1 and Oligo_2 remain close due to similar gene expression, while other cell types are more distinct. The observed shifts in cell-type proportions between groups provide insights into AD-related changes and algorithmic performance. In comparison, GE-Impute underestimates cluster numbers, sometimes merging biologically distinct subtypes and missing genuine heterogeneity.

Fig. 8 indicate significant changes in the proportion of different cell types. The number of Oligo_3, astrocytes, and oligodendrocyte precursor cells has decreased, while the number of Oligo_1, Oligo_2, and Oligo_4 has increased. Recent studies Kedia & Simons (2025) in AD have revealed the reasons for these changes.

### 3.2.4 Ablation Study.

The scCapsZB model combines a ZINB autoencoder and a capsule network. Two conditions were tested: only-ZINB and only-Caps, both using the same preprocessing. Fig. 9 shows the results of these tests. scCapsZB performed best on the 10X_PBMC, Romanov, Zeisel, and Klein datasets. On the Human1 dataset, scCapsZB and only-ZINB had similar results, while only-Caps performed the worst due to missing diverse gene cell data. On the Human2 dataset, only-Caps performed the best, and only-ZINB performed the worst because the dataset didn't fit the ZINB model well. Overall combining the capsule network and the ZINB autoencoder improved clustering performance, showing the validity and effectiveness of all its components.

### 3.3 KEGG Pathway Enrichment Analysis

In the R environment, the ClusterProfiler package was directly downloaded and used to perform KEGG enrichment analysis on the 2000 highly differentially expressed genes retained after imputation in AD cells. The gene pathways were sorted based on the enrichment factor, and the top 30 pathways were selected for visualization, as shown in Figure 10. The biological pathway enrichment analysis revealed that several pathways were highly positively enriched in AD cells, including Sphingolipid metabolism, Alzheimer disease, and Huntington disease-related pathways, all of which exhibited significant p-values, high enrichment factors, and a large aggregation of genes. These pathways represent the pathogenic routes that contribute to neural degeneration and even mortality in patients; thus, further research on the genes within these pathways could help elucidate the pathological mechanisms underlying Alzheimer's and other neurodegenerative diseases. Notably, due to recent extensive research on COVID-19 gene pathways and updates to the KEGG database, the Coronavirus disease (COVID-19) pathway was also significantly enriched in AD patients—likely reflecting the advanced age of these patients, which makes them more susceptible to COVID-19 and results in pronounced related gene expression characteristics.

## 4 Conclusion

Across multiple public datasets and an Alzheimer's disease cohort, our framework consistently improves imputation fidelity and clustering accuracy over existing state-of-the-art approaches, demonstrating its robustness across different tissue types, sequencing depths, and experimental conditions. The high-quality embeddings produced by our method enable not only the recovery of biologically meaningful cell states but also the detection of rare or previously overlooked subpopulations that may play critical roles in neurodegeneration. In particular, our analysis highlights transcriptional programs related to immune activation, synaptic dysfunction, and glial reactivity, which are highly relevant to Alzheimer's disease pathology and may represent potential therapeutic targets. These findings underscore the importance of integrating graph-based representation learning with biologically informed clustering to better resolve cellular heterogeneity in complex tissues. Furthermore, quantitative benchmarking shows that our approach achieves lower reconstruction error and higher adjusted Rand index than competing methods, even on noisy and sparse datasets, confirming its generalizability. The modular nature of our pipeline allows it to be easily extended to multimodal single-cell data, such as scATAC-seq or spatial transcriptomics, paving the way for comprehensive cross-omic integration and deeper mechanistic insights into disease progression.

Together, scICLGAE and scCapsZB form a robust and extensible pipeline for denoising, embedding, and interpreting large-scale single-cell datasets. The modular design of the framework allows seamless integration with downstream analyses such as trajectory inference, differential expression testing, and cell–cell communication modeling, thereby broadening its applicability to diverse biological questions. We anticipate that this work will facilitate mechanistic discovery and biomarker development for Alzheimer's disease and other neurodegenerative disorders, and we acknowledge the use of ChatGPT for language polishing and refinement during manuscript preparation.

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

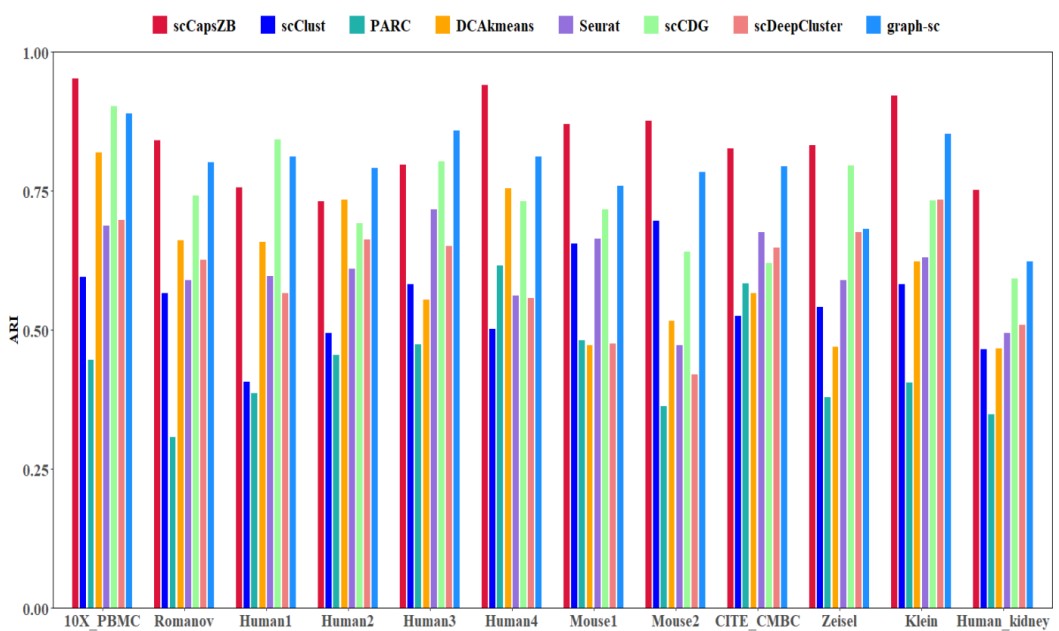

Figure 5: Comparison of ARI metrics.

## A APPENDIX

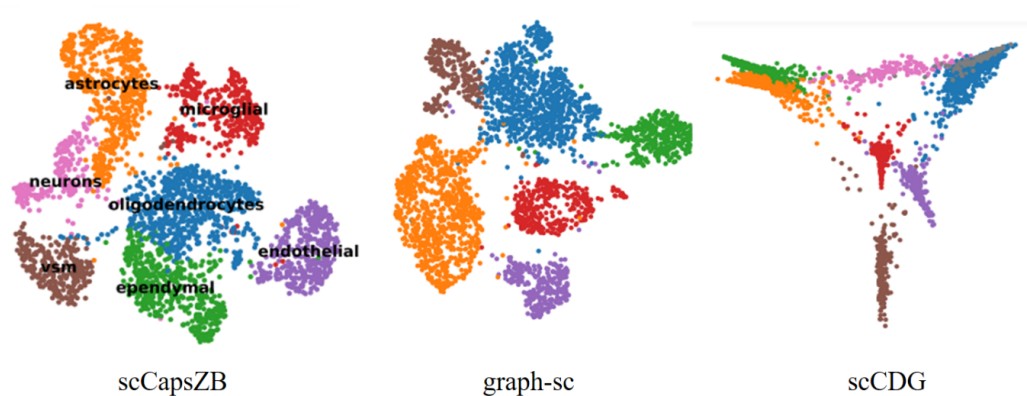

scCapsZB                graph-sc                scCDG

Figure 6: Visualization result comparison between scCapsZB and two graph neural network algorithms.

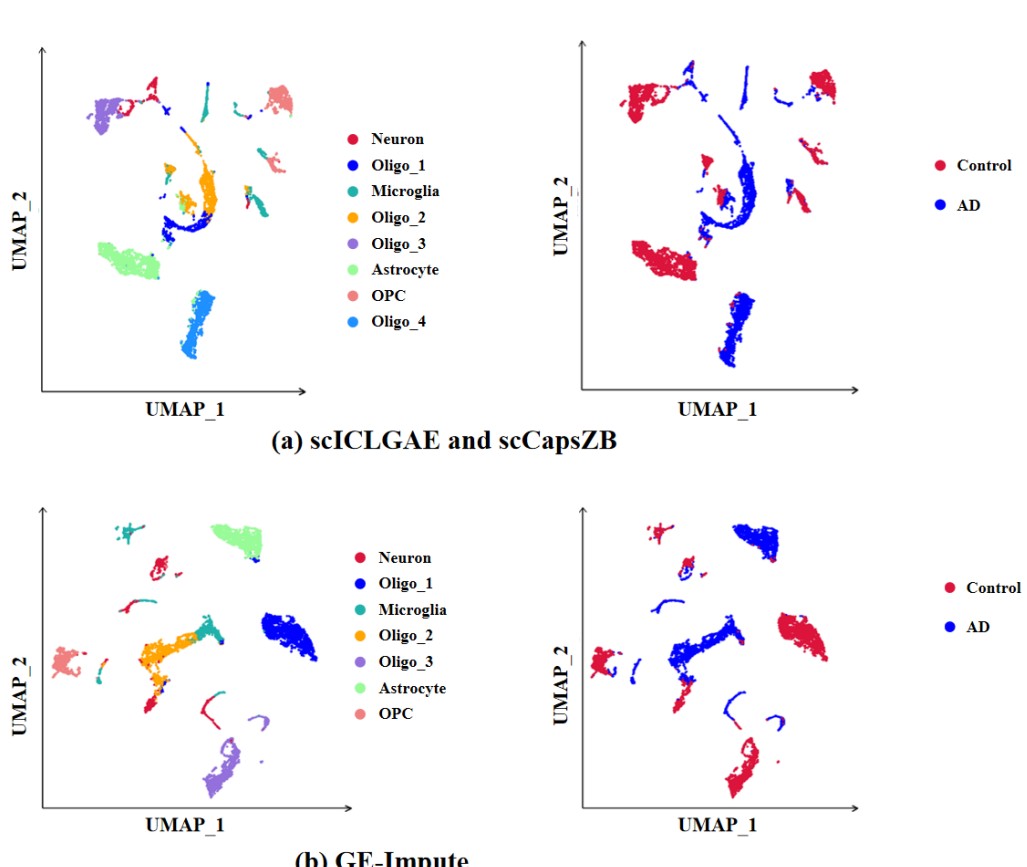

Figure 7: Distribution of imputation and clustering results in the control and patient group.

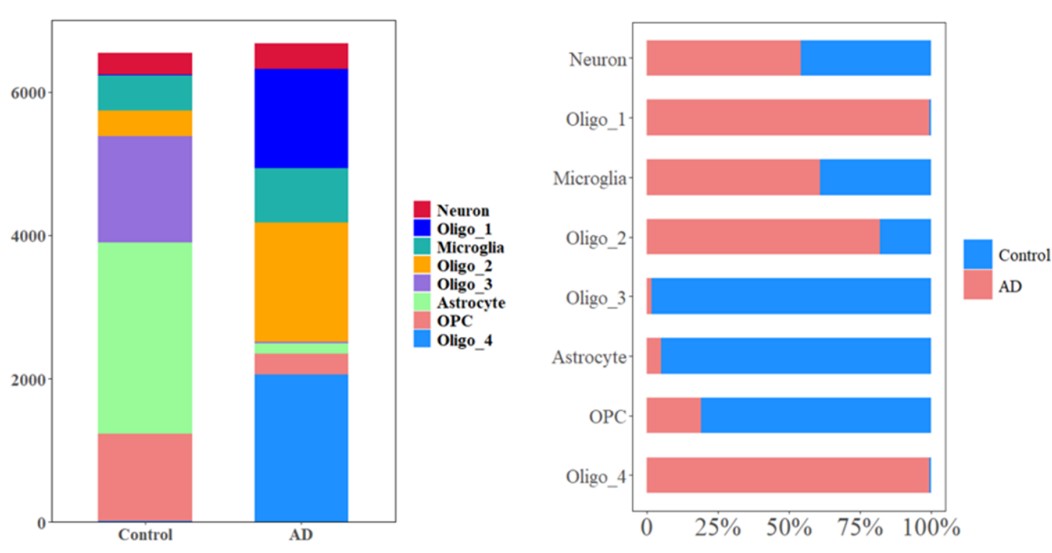

Figure 8: Comparison of the proportions of various cell numbers in the AD patient group and the control group.

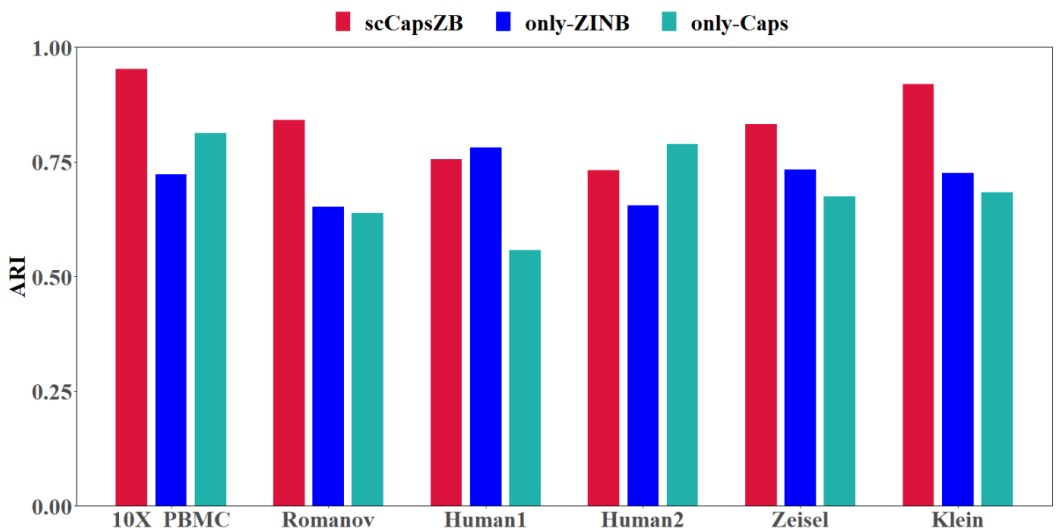

Figure 9: The results of ablation experiments.

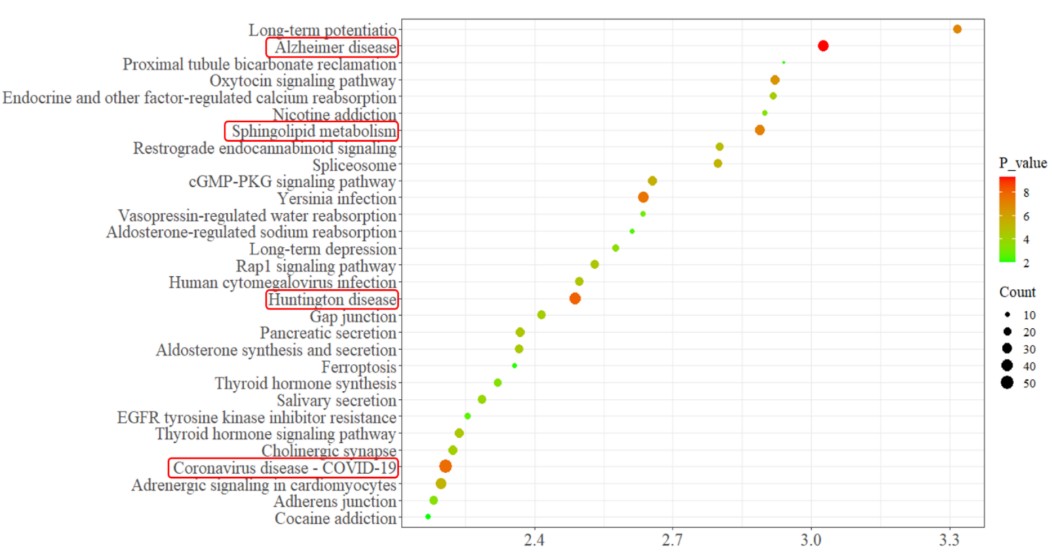

Figure 10: The results of KEGG Analysis.

