# OpenReview forum: "Alzheimer's scRNA-seq Data Analysis Using Multi-type Deep Autoencoders"
_ICLR.cc/2026/Conference — Submitted to ICLR 2026_

### Official Review · Reviewer_PSX9 · 2025-10-29

**Soundness:** 2
**Presentation:** 2
**Contribution:** 1
**Rating:** 2
**Confidence:** 4

**Summary:**

Summary
The paper proposes a two-stage scRNA-seq pipeline: an imputation module that builds a cosine-KNN graph and contrastive graph autoencoder to fill zeros, followed by a capsule-network clustering module coupled to a ZINB autoencoder with a DEC-style target distribution and a small set of true labels. It reports better imputation/clustering on multiple datasets and presents an Alzheimer’s case study with cell-proportion shifts and KEGG enrichments. Overall, the method reads as an incremental assembly of known components, with key baselines missing.

**Strengths:**

Strength:
1. Clear modularity (imputation → clustering) makes the system easy to implement and ablate conceptually.
2. Sensible choice of ZINB likelihood for count data (if used on raw counts) and a DEC-style self-training loop to refine cluster assignments.
3. Practical focus on Alzheimer’s data provides an application context beyond toy benchmarks.

**Weaknesses:**

Weakness:
1. The method largely assemble established ingredients (contrastive graph AE, DEC target sharpening, ZINB AE, capsule routing) without a new principle, analysis, or guarantee.
2. The clustering stage uses ground-truth labels (semi-supervised) while baselines are unsupervised; there’s no head-to-head with scANVI or other label-aware methods under the same label budget.
3. Widely used scRNA-seq VAEs/autoencoders (scVI/LDVAE/SAUCIE, etc.) are cited but not included, undermining any SOTA claim.
4. Cosine KNN in imputation and Pearson KNN in clustering are used without justification; it’s also unspecified whether ZINB is fit to raw counts, size-factor–normalized counts, or log space—crucial for model validity.
5. The Alzheimer’s analysis lacks subject-level statistics: proportion shifts and DE appear pooled across cells (risking pseudoreplication), and pathway enrichment is run on DE from imputed matrices, which can inflate false positives.
6. Key hyperparameters are missing on key knobs: KNN size, Louvain resolution, labeled-data fraction, capsule hyperparameters, neighborhood size for imputation, random seeds; no runtime/memory reporting.
7. Claims of multimodal/general applicability (scATAC, spatial) are made without any cross-omic evidence.
8. Reporting and reproducibility are under-specified (preprocessing, masking, label ratio, early stopping, compute details), and figures lack error bars and statistical tests.

**Questions:**

No additional questions.

**Details Of Ethics Concerns:**

No ethics concerns.

---

### Official Review · Reviewer_Tz5J · 2025-10-31

**Soundness:** 2
**Presentation:** 2
**Contribution:** 3
**Rating:** 2
**Confidence:** 3

**Summary:**

The paper proposes a two-stage pipeline: scICLGAE for imputation (dual GAEs with contrastive learning that pull local KNN and global Louvain neighbors) and scCapsZB for clustering (a ZINB autoencoder coupled with a capsule network using self-attention routing).

**Strengths:**

The architecture is well aligned with the task: scICLGAE explicitly fuses local and global cell neighborhoods before imputation, while scCapsZB models count noise with ZINB and leverages capsule routing to capture hierarchical structure.

**Weaknesses:**

1. Beyond the three simulated rates (10/30/50%), the evaluation should include extreme dropout levels (e.g., 80% or 90%) to reflect low-coverage cells and rare populations.
2. Baseline coverage is incomplete. Imputation is compared to only four methods (DeepImpute, scIGANs, GNNImpute, GE-Impute), omitting non-graph (e.g., DCA [1], AutoClass[2]), GNN-based (e.g., scTAG[3], scGNN 2.0[4], scGCL[5]), and propagation-based (e.g., MAGIC[6], scFP[7], scBFP[8], scCR[9]). It provides no references or experimental comparisons for these families.
3. Ablations are insufficient to pinpoint where the gains come from.

[1] Gökcen Eraslan, Lukas M Simon, Maria Mircea, Nikola S Mueller, and Fabian J Theis. Singlecell rna-seq denoising using a deep count autoencoder. Nature communications, 10(1):390, 2019.

[2] Hui Li, Cory R Brouwer, and Weijun Luo. A universal deep neural network for in-depth cleaning of single-cell rna-seq data. Nature Communications, 13(1):1901, 2022.

[3] Zhuohan Yu, Yifu Lu, Yunhe Wang, Fan Tang, Ka-Chun Wong, and Xiangtao Li. Zinb-based graph embedding autoencoder for single-cell rna-seq interpretations. In Proceedings of the AAAI conference on artificial intelligence, volume 36, pages 4671–4679, 2022.

[4] Juexin Wang, Anjun Ma, Yuzhou Chang, Jianting Gong, Yuexu Jiang, Ren Qi, Cankun Wang, Hongjun Fu, Qin Ma, and Dong Xu. scgnn is a novel graph neural network framework for single-cell rna-seq analyses. Nature communications, 12(1):1882, 2021.

[5] Zehao Xiong, Jiawei Luo, Wanwan Shi, Ying Liu, Zhongyuan Xu, and Bo Wang. scgcl: an imputation method for scrna-seq data based on graph contrastive learning. Bioinformatics, 39(3):btad098, 2023.

[6] David Van Dijk, Roshan Sharma, Juozas Nainys, Kristina Yim, Pooja Kathail, Ambrose J Carr, Cassandra Burdziak, Kevin R Moon, Christine L Chaffer, Diwakar Pattabiraman, et al. Recovering gene interactions from single-cell data using data diffusion. Cell, 174(3):716–729, 2018.

[7] Sukwon Yun, Junseok Lee, and Chanyoung Park. Single-cell rna-seq data imputation using feature propagation. arXiv preprint arXiv:2307.10037, 2023.

[8] Junseok Lee, Sukwon Yun, Yeongmin Kim, Tianlong Chen, Manolis Kellis, and Chanyoung Park. Single-cell rna sequencing data imputation using bi-level feature propagation. Briefings in Bioinformatics, 25(3):bbae209, 2024.

[9] Daeho Um, Ji Won Yoon, Seong Jin Ahn, and Yunha Yeo. Gene-Gene Relationship Modeling Based on Genetic Evidence for Single-Cell RNA-Seq Data Imputation. in Proc. NeurIPS, 2024.

**Questions:**

Please refer to the concerns outlined in the Weaknesses section.

---

### Official Review · Reviewer_ZW6G · 2025-11-01

**Soundness:** 2
**Presentation:** 1
**Contribution:** 1
**Rating:** 2
**Confidence:** 4

**Summary:**

This paper addresses the problem of imputing and clustering single cell RNA sequence data of Alzheimer’s disease.  It proposes two models: scICLGAE which is a graph autoencoder trained with contrastive learning loss on local and global neighbors for imputation, and scCapsZB which is a Zero-Inflated Negative Binomial autoencoder with a self-attention capsule network for clustering.

**Strengths:**

-scICLGAE combines local graph structure (from KNN) with global structure (from Louvain clustering on GAE embeddings), which is a good heuristic for finding a robust set of similar cells to use for imputation.
-The ablation study (Figure 9) provides good evidence that the combination of the ZINB autoencoder and the capsule network in scCapsZB is beneficial and that the full model generally outperforms the individual components.
-The application to a real, complex Alzheimer's disease dataset and the subsequent downstream KEGG analysis is a nice addition, showing the contribution in a relevant biological problem.

**Weaknesses:**

The problem this paper tackles is well-addressed with a large number of existing methods [5-8], which diminishes its significance. In order to demonstrate the advantage of the method, it would need to outperform existing methods on an extensive benchmark such as openproblems.bio [8].
The paper main uses graph/GNN, which is almost used in all methods for single cell data, limiting its novelty [1, 4-7, 9-12].
The paper introduces overcomplicated frameworks without any motivation for those frameworks. For example, it uses contrastive learning, but did not justify why it would be helpful in the imputation tasks. Similar for the self-attention routing capsule network module. This unnecessary complication leads to concerns in robustness and computational efficiency.
The paper did not compare with the most highly-cited single cell methods like MAGIC[1], scImpute[2] and SAVER[3] for imputation, and Leiden[4] for clustering.
The method uses cosine similarity on raw sequence counts, but it is known that cosine similarity is going to suffer for high dimensionality and large noise, and the computational cost is very high given the high dimensionality.
The paper claims to be focused on Single-cell RNA sequencing data of Alzheimer’s Disease, but there is no specific method design that utilized the properties of AD data, making it unclear why it would have benefit on such data, compared to scRNAseq data in general.
For comparison with baselines, there is no error bar or uncertainty measure to show the statistical significance. Moreover, the proposed method does not outperform the existing baselines overall.
The paper has some issues in clarity and presentation.
Comparison results are not shown in a clear table but are in nonstandard plots. Besides, there is a global mismatch between the figure/equation numbers and their reference.
There also needs some explanations to the evaluation metric. The notations of $$y_i$$ and $$p_k$$ are also ambiguous: they are described as cell indices in the text but probably are the expression values.
The prior probability matrix $$L^i$$ being used is not defined in the text.
The negative sample generator in Figure 2 is also not well defined.
Equation numbers are misaligned and/or missing. Equation 23 is referred to as equation 24.
The pathway analysis results are overinterpreted. For example, the conclusion about the implication of COVID pathways in the AD dataset is not sufficiently justified.

References:

[1] Van Dijk, David, et al. "Recovering gene interactions from single-cell data using data diffusion." Cell 174.3 (2018): 716-729.
[2] Li, Wei Vivian, and Jingyi Jessica Li. "An accurate and robust imputation method scImpute for single-cell RNA-seq data." Nature communications 9.1 (2018): 997.
[3] Huang, Mo, et al. "SAVER: gene expression recovery for single-cell RNA sequencing." Nature methods 15.7 (2018): 539-542.
[4] Traag, Vincent A., Ludo Waltman, and Nees Jan Van Eck. "From Louvain to Leiden: guaranteeing well-connected communities." Scientific reports 9.1 (2019): 1-12.
[5] Cheng, Yi, et al. "Evaluating imputation methods for single-cell RNA-seq data." BMC bioinformatics 24.1 (2023): 302.
[6] Hou, Wenpin, et al. "A systematic evaluation of single-cell RNA-sequencing imputation methods." Genome biology 21.1 (2020): 218.
[7] Zhang, Shixiong, et al. "Review of single-cell RNA-seq data clustering for cell-type identification and characterization." Rna 29.5 (2023): 517-530.
[8] Luecken, Malte D., et al. "Defining and benchmarking open problems in single-cell analysis." Nature Biotechnology (2025): 1-6.
[9] Blondel, Vincent D., et al. "Fast unfolding of communities in large networks." Journal of statistical mechanics: theory and experiment 2008.10 (2008): P10008.
[10] Levine, Jacob H., et al. "Data-driven phenotypic dissection of AML reveals progenitor-like cells that correlate with prognosis." Cell 162.1 (2015): 184-197.
[11] Song, Qianqian, Jing Su, and Wei Zhang. "scGCN is a graph convolutional networks algorithm for knowledge transfer in single cell omics." Nature communications 12.1 (2021): 3826.
[12] Weinreb, Caleb, Samuel Wolock, and Allon M. Klein. "SPRING: a kinetic interface for visualizing high dimensional single-cell expression data." Bioinformatics 34.7 (2018): 1246-1248.

**Questions:**

What is the key advance over the dozens of methods offered for these oft-tackled problems in single cell analysis?

---

### Official Review · Reviewer_5LZn · 2025-11-02

**Soundness:** 3
**Presentation:** 3
**Contribution:** 2
**Rating:** 4
**Confidence:** 4

**Summary:**

The paper introduces a novel two-stage framework for analyzing single-cell RNA sequencing (scRNA-seq) data in Alzheimer’s disease (AD) using deep learning. It integrates: scICLGAE, a graph contrastive learning-based imputation algorithm using dual graph autoencoders to address missing data and denoise gene expression; scCapsZB, a clustering algorithm combining a Zero-Inflated Negative Binomial (ZINB) autoencoder and a capsule network with self-attention routing and semi-supervised learning. The framework was tested on 12 scRNA-seq datasets and an Alzheimer’s dataset, showing improved imputation accuracy and clustering performance compared to existing methods.

**Strengths:**

1. Innovative hybrid model design: Combines contrastive graph learning, ZINB-based feature modeling, and capsule networks, which is a fresh integration in scRNA-seq analysis.
2. Comprehensive methodology: The paper details every computational component (graph construction, loss functions, semi-supervised learning, etc.) with equations and workflow illustrations.
3. Strong benchmarking: Tested on a large set of public datasets covering various sequencing platforms and biological contexts. Provides comparative evaluations with multiple state-of-the-art methods

**Weaknesses:**

1. Complexity and reproducibility: The framework’s multiple interconnected modules (contrastive learning, capsule routing, ZINB modeling) may pose challenges for reproducibility and computational efficiency.
2. Limited biological interpretation: While KEGG analysis is provided, the biological implications of specific gene or cell-type findings in AD are not deeply discussed.
3. Dataset scope: Only one AD dataset was used for disease-specific evaluation; broader validation (e.g., multiple brain regions, independent cohorts) would strengthen conclusions.
4. Quantitative results: Figures show visual trends, but numerical results (e.g., mean ± SD) are not tabulated, which limits quantitative reproducibility.

**Questions:**

1. How sensitive are the results to hyperparameter choices, such as the KNN size, capsule dimensions, or ZINB parameters?
2. Is there any biological validation (e.g., wet-lab or literature-based confirmation) for the newly identified AD-related cell-type shifts?
3. How does the capsule module handle imbalanced or sparse label availability?

---

### Meta-Review · Area_Chair_hb4a · 2026-01-05

**Summary:**

Reviewers are all negative about this work due to some major issues.

**Reviewer Scores:**

n/a

---

### Decision · Program_Chairs · 2026-01-26

Reject